# Biodistribution and Pharmacokinectics of Liposomes and Exosomes in a Mouse Model of Sepsis

**DOI:** 10.3390/pharmaceutics13030427

**Published:** 2021-03-22

**Authors:** Amin Mirzaaghasi, Yunho Han, So-Hee Ahn, Chulhee Choi, Ji-Ho Park

**Affiliations:** 1Department of Bio and Brain Engineering and KAIST Institute for Health Science and Technology, Korea Advanced Institute of Science and Technology (KAIST), Daejeon 34141, Korea; amaghasi@kaist.ac.kr (A.M.); patientyh@kaist.ac.kr (Y.H.); 2Analytic Development Team, ILIAS Biologics Incorporated, Daejeon 34014, Korea; shahn@iliasbio.com

**Keywords:** biodistribution, exosome, liposome, sepsis, pharmacokinetics, near-infrared imaging

## Abstract

Exosomes have attracted considerable attention as drug delivery vehicles because their biological properties can be utilized for selective delivery of therapeutic cargoes to disease sites. In this context, analysis of the in vivo behaviors of exosomes in a diseased state is required to maximize their therapeutic potential as drug delivery vehicles. In this study, we investigated biodistribution and pharmacokinetics of HEK293T cell-derived exosomes and PEGylated liposomes, their synthetic counterparts, into healthy and sepsis mice. We found that biodistribution and pharmacokinetics of exosomes were significantly affected by pathophysiological conditions of sepsis compared to those of liposomes. In the sepsis mice, a substantial number of exosomes were found in the lung after intravenous injection, and their prolonged blood residence was observed due to the liver dysfunction. However, liposomes did not show such sepsis-specific effects significantly. These results demonstrate that exosome-based therapeutics can be developed to manage sepsis and septic shock by virtue of their sepsis-specific in vivo behaviors.

## 1. Introduction

Exosomes have garnered much interests in recent drug delivery systems (DDSs) because they have been identified as potential vehicles for transmitting various types of therapeutic agents to target cells [1,2,3]. Due to their nature as an intercellular communicator and their nanoscale size in the range of 40–150 nm, exosomes have been implicated in many pharmaceutical applications [1,4,5,6,7]. For example, exosomes derived from mesenchymal stem cells (MSC) have been extensively studied as therapeutic agents for cancer treatment and tissue regeneration, and there are a few studies in clinical trials [5,8,9,10,11,12]. In addition, exosomes in biological fluids including blood, urine, saliva, and breast milk have been used as biomarkers for disease diagnosis because they carry the information of their parental cells [13,14,15,16].

Among all nanoscale formulations used in DDSs, liposomes have gained valuable results for many years due to their biocompatibility and ability to entrap various drugs in their internal core and membrane [17,18,19]. In addition, the liposomal surface can be easily modified with various functional molecules, including targeting ligands and polyethylene glycol (PEG) with stealth properties, which has led to successful therapeutic applications of liposomes in many diseases. In vivo behavior of liposomes has shown that they can efficiently escape the mononuclear phagocytic system (MPS) and reside relatively long in the blood stream [20,21,22,23,24]. After a prolonged period of time, systemically-administered liposomes accumulate mainly in the liver and spleen.

Recent advances in DDS have enabled the improvement of treatment outcomes in many life-threatening conditions. Sepsis is the condition caused by the extreme immune response to an infection and is associated with high mortality rates in intensive care units. There are currently very few options for the treatment of sepsis, and they mainly include antibiotics and intravenous fluids, and their therapeutic results have been limited due to uncontrolled systemic inflammation. Recently, liposomes and exosomes in the DDS have been engineered to treat sepsis and septic shock. Liposomes have been used to improve delivery efficacy of antibiotics to inflammation regions and protect mice against invasive infections by sequestering bacterial endotoxin on their membranes [25]. Exosomes derived from various types of cells have been engineered to deliver therapeutic RNAs or proteins to target cells for the reduction of inflammation in sepsis [26,27,28,29]. Furthermore, it was recently reported that intraperitoneal injection of exosomes loaded with srIκB significantly reduced mortality rate and organ damage by inhibiting NF-κB activity in the inflammatory regions [30]. Although these studies suggested that, similar to liposomes, exosomes have great potential as drug delivery vehicles for the management of sepsis, the limited number of studies examining their pharmacokinetics and biodistribution in the sepsis model made it hard to understand the mechanism of action behind their therapeutic effects.

Here, we examine in vivo pharmacokinetics and biodistribution of liposomes and exosomes in a mouse model of sepsis. Liposomes and exosomes are labeled with a near-infrared (NIR) lipophilic dye, DiR, for whole-body and ex vivo organ fluorescence imaging. DiR-labeled liposomes and exosomes are intravenously injected into both healthy and sepsis mice, and the time-lapse fluorescence images of whole-body and ex vivo organs, including the liver, spleen, lung, brain, kidney, GI tract, and blood, were analyzed to understand their disease-specific pharmacokinetics and biodistribution.

## 2. Materials and Methods

### 2.1. Cell Cultures

HEK293T human kidney cells (CRL-3216, ATCC, Manassas, VA, USA) were cultured in Dulbecco’s modified Eagle’s medium (DMEM; Welgene, Seoul, Korea) containing 10% fetal bovine serum (FBS; Gibco, Gaithersburg, MD, USA) and 1% penicillin–streptomycin (Gibco).

### 2.2. Exosome Purification

HEK293T cell-derived exosomes were purified as previously described [30]. In brief, HEK293T cells were plated on a T175 flask for 24 h, washed with PBS, and cultured in medium containing 10% exosome-depleted FBS for 72 h. The medium was collected and centrifuged at 1000× *g* for 15 min to remove cells and cell debris. A 0.22-µm polyethersulfone filter was used to remove large particles. Exosomes were isolated by molecular weight cutoff-based membrane filtration and run through size exclusion chromatography for purification.

### 2.3. Preparation of DiR-Labeled Liposomes

Hydrogenated soy phosphatidylcholine (HSPC; Avanti Polar Lipids, Alabaster, AL, USA), cholesterol (Chol; Avanti Polar Lipids), 1,2-distearoyl-sn-glycero-3-phosphoethanolamine-*N*-(methoxy(polyethylene glycol)-2000) (ammonium salt) (DSPE-PEG; Avanti Polar Lipids), and near-infrared fluorescent lipophilic dye, 1,1′-dioctadecyl-3,3,3′,3′tetramethylindotri carbocyanine iodide (DiR; Cat# D12731, Thermo Fisher, Waltham, MA, USA) were used to prepare DiR-labeled liposomes. HSPC, Chol, and DSPE-PEG were dissolved in chloroform in the molar ratio of 56:39:5 (HSPC:Chol:DSPE-PEG) with a total lipid mole number of 1.41 × 10^−6^. The DiR was dissolved with 1 mg/mL in ethanol; 10 µL of a DiR stock solution was added to the lipid mixture solution and incubated at room temperature overnight to dry out. The remaining DiR-embedded lipid film was mixed with 1 mL of phosphate-buffered saline (PBS), and the mixture solution was extruded through 50-nm membrane pores (Whatman, Little Chalfont, UK). The distribution of hydrodynamic sizes was measured by dynamic light scattering (Zetasizer Nano ZS90; Malvern Instruments, Malvern, UK).

### 2.4. Western Blot

Western blot was performed based on the procedure described by Yim et al. [3]. Antibody solutions were prepared for CD9 (1:500 dilution; Cat# EXOAB-CD9A-1, SBI), Tsg101 (1:1000 dilution; Cat# ab83, Abcam, Cambridge, UK), GM130 (1:1000 dilution; Cat# Ab52649, Abcam), and GAPDH (1:4000 dilution; Cat# sc-47724, Santa Cruz Biotechnology, Dallas, TX, USA).

### 2.5. Nanoparticle Analysis

Nanoparticle tracking analysis (NTA) was performed using a Zetaview^®^ simulator (Cat#PMX120, Particle Metrix, Inning am Ammersee, Germany). All samples were diluted between 1:100 and 1:10,000 for measurements. The particle number video was captured based on Brownian motion, and the hydrodynamic particle diameter was determined based on the two-dimensional Stokes–Einstein equation. The particle count was measured between 50 and 200 particles/frame.

### 2.6. Preparation of DiR or DiO-Labeled Exosomes

For DiR or 3,3′-dioctadecyloxacarbocyanine perchlorate (DiO; Cat# D275, Thermo Fisher) labeling, the purified exosomes were incubated with 50 µg/mL DiR or DiO solution at a ratio of 2% (*v*/*v*) at 37 °C overnight and then loaded on the size-ex­clusion chromatography column (G-25 M Sephadex GE Healthcare, Little Chalfont, UK). Eluted fractions were collected dropwise on a UV 96-well plate, and the absorbance of the fractions was measured at the wavelength range of 200–400 nm to identify the protein peak at 280 nm. The fractions with a peak in the absorbance at 280 nm were transferred to a centrifugal filter tube (Cat# UFC810024, Amicon, Miami, FL, USA) and centrifuged at 5000× *g* for 30 min at 4 °C. After centrifugation, the DiR- or DiO-labeled exosomes were resuspended in PBS.

### 2.7. TEM

Five microliters of HEK293T cell-derived exosomes suspended in PBS was applied onto glow-discharged carbon-coated copper grids for around 5 s. The samples on the grids were each stained with 2% uranyl acetate and then blotted with a filter paper. The samples were air-dried at room temperature and observed with Tecnai G2 Retrofit (FEI Company, Hillsboro, OR, USA) at a 200 kV voltage.

### 2.8. Animal Model

Eight-week-old female C57BL/6 mice from Orientbio (Seong-Nam, Korea) were used. A mouse model of sepsis was prepared by intravenously injecting mice with a single dose of 30 µg/kg lipopolysaccharide (LPS) derived from *Escherichia coli* (Sigma-Aldrich, Milwaukee, WI, USA). For all experiments, liposomes or exosomes were intravenously injected 30 min after LPS treatment. Mice were anesthetized through intraperitoneal injection of Zoletil 50 (25 mg/kg) and Rompun (1:1) prior to all procedures of organ dissection and blood collection. All animal experiment protocols were approved by the Institutional Animal Care and Use Committee (IACUC) of KAIST (Korea Advanced Institute of Science and Technology).

### 2.9. Fluorescence Imaging

For whole body imaging, mice received an intravenous injection of DiR-labeled exosomes at a particle number of 3 × 10^9^/injection or DiR-labeled liposomes at a particle number of 2.8 × 10^9^/injection. Representative whole-body images of the mice were taken with VISQUE (VISQUE inVivo Elite, Viewworks, Anyang-si, Korea). The mice were imaged at excitation and emission wavelengths of 750 nm and 810 nm, respectively, under anesthesia at 5 min, 1 h, and 2 h after exosome or liposome injection. The intensity of DiR fluorescence in liver, lung, and spleen was quantified by using Viewworks software. For blood imaging, 150 to 200 µL of blood samples were collected by retro-orbital puncture of the mice under anesthesia at 30 min, 1 h, 2 h, 4 h, 8 h, and 24 h after exosome or liposome injection. For organ imaging, liver, spleen, lung, brain, kidney, and GI tract were harvested from the sacrificed mice at 30 min, 1 h, 2 h, 4 h, 8 h, and 24 h after exosome or liposome injection. Organs and blood samples were then imaged using an Odyssey imaging system (LI-Core, Lincoln, NE, USA), and the images were analyzed with Image Studio 4.021 software. The blood half-life was calculated by fitting the fluorescence intensity measured at each time point to a one-compartment open pharmacokinetic model. Samples were taken from five mice at each time point.

### 2.10. Confocal Microscopy

To examine cellular uptake of exosomes in the lung tissues, mice received an intravenous injection of DiO-labeled exosomes at a particle number of 3 × 10^9^/injection. One hour after injection, mice were sacrificed and lungs were harvested. Dissected lungs were embedded in OCT compound (Leica, Wetzlar, Germany) and frozen at −70 °C. Frozen samples were sectioned with 10 µm of thickness using a freezing microtome (Leica CM3050 S; Leica Biosystems, Wetzlar, Germany). For macrophage staining, the lung sections were incubated with Alexa Fluor 647 anti-mouse F4/80 antibody (1:50 dilution with PBS, Invitrogen, Carlsbad, CA, USA) for one hour and washed with PBS. For nucleus staining, the lung sections were incubated with Hoechst 33342 (2 µg/mL in PBS, Sigma) for 15 min at room temperature and washed with PBS. The stained lung sections were then observed using confocal microscopy (Nikon, Tokyo, Japan).

## 3. Results and Discussion

We first prepared liposomes and exosomes labeled with a NIR fluorescence dye to examine their biodistribution and pharmacokinetics using NIR fluorescence imaging. Purified HEK293T cell-derived exosomes were used for all following experiments. TEM of HEK293T cell-derived exosomes revealed that they appeared spherical shaped with an average size of 90 nm (Figure 1A). Western blotting analysis confirmed that CD9 and TSG101, two common exosome markers, were present in the exosomes, while GM130, Golgi-derived contaminant, was only observed in the cell lysate (Figure 1B). DiR, a lipophilic NIR fluorescence dye that can be loaded in the lipid bilayer, was used for labeling both liposomes and exosomes. Mean sizes of DiR-loaded liposomes and exosomes were approximately 80 nm and 100 nm, respectively, which was slightly increased from their original sizes after DiR labeling (Figure 1C,D). Since free DiR molecules can adhere to plasma proteins with hydrophobic domains, DiR-loaded liposomes and exosomes were purified extensively using the size exclusion chromatography (SEC) method (Appendix A).

The toxic issues of LPS are mediated by cytokines like TNF-α, interleukin 1 (IL-1), and IL-6 released by LPS-activated host cells, such as neutrophils and macrophages. Several studies have shown that initial pro-inflammatory cytokines increase rapidly within 30 min after LPS injection [31,32]. Our recent cellular biodistribution findings also confirm the significant therapeutic effect of exosome carrying srIKB during the early stage of sepsis before the immunosuppressive state [30]. Therefore, in this study, exosomes and liposomes were injected 30 min after LPS injection to investigate their biodistribution in the early stage of sepsis. Since efficient tissue penetration of NIR fluorescence allows for real-time monitoring of the fluorescence in living mice, we next carried out time-lapse whole-body NIR fluorescence imaging of mice after intravenous injection of DiR-loaded liposomes or exosomes. Whole-body fluorescence images showed that liposomes accumulated more slowly in the liver compared to exosomes (Figure 2), indicating that polyethylene glycol (PEG) moieties on the liposomal surface are likely to delay the clearance by Kupffer cells, macrophages resident in the liver. Liposomes accumulated mainly in the healthy liver over the time course after intravenous injection, while a substantial amount translocated from the liver to the intestine in sepsis mice likely from 3 h post-injection. Exosomes accumulated mainly in the liver in both healthy and sepsis mice within 1 h after intravenous injection. Interestingly, exosomes accumulated in the healthy liver translocated to the intestine from 2 h post-injection and disappeared over time, while the sepsis liver somewhat delayed translocation of exosomes to the intestine. These results suggest that liposomes and exosomes have different biodistribution and pharmacokinetics in both healthy and sepsis mice.

To examine time-dependent accumulation of liposomes and exosomes in major organs, we next performed NIR fluorescence imaging of ex vivo organs (liver, brain, heart, lung, spleen, kidney, and intestine) obtained at different time points after intravenous injection. Similar to what was observed in the whole-body imaging, systemically-administered liposomes and exosomes were found mainly in the liver and spleen of both healthy and sepsis mice (Figure 3), which are major organs in the mononuclear phagocyte system (MPS). Exosomes cleared in the healthy liver were translocated to the intestine from 8 h post-injection, while exosomes injected into the sepsis mice did not show such translocation (Figure 3A). These results imply that liver dysfunction in a later stage of sepsis may delay biliary excretion of exosomes. On the other hand, the covalent linking of polyethylene glycol (PEG) chains is an effective method to prolong the blood circulation time of liposomes. The resistance of protein opsonin absorbance onto liposome surfaces caused by PEGylation delayed their elimination by the mononuclear phagocytic cells in the liver and spleen (Figure 3B). Importantly, we found that a substantial number of exosomes accumulated in the lung of sepsis mice over the time course after intravenous injection, implying that the immune cells involved in acute lung injury (ALI) in sepsis, such as macrophages and neutrophils, may specifically take up exosomes in the blood circulation (Figure 3C). The confocal microscopy of lung tissue sections also confirmed the effective uptake of exosomes by macrophages increased in the inflamed lungs (Appendix A). Liposomes did not show sepsis-specific accumulation in the lung (Figure 3B,D). Lastly, we evaluated the pharmacokinetics of liposomes and exosomes in the blood of healthy and sepsis mice. In the healthy mice, over 80% of exosomes was cleared from blood circulation within 1 h after intravenous injection (Figure 4A). The blood clearance of exosomes was remarkably delayed in the sepsis mice (Figure 4B), re-confirming the liver dysfunction in sepsis. However, the blood residence of liposomes was not affected significantly by sepsis conditions. Collectively, these results suggest that cell-derived exosomes seem to be more affected in the blood by pathophysiological conditions of sepsis than synthetic liposomes.

## 4. Conclusions

In this study, we found that biodistribution and pharmacokinetics of HEK293T cell-derived exosomes were significantly affected by pathophysiological conditions of sepsis compared to those of PEGylated liposomes. In the sepsis mice, a substantial number of exosomes were found in the lung after intravenous injection, and their prolonged blood residence was observed due to liver dysfunction. However, liposomes did not show such sepsis-specific effects significantly. Thus, these findings might help in better understanding of the exosome and liposome organ distribution, particularly in the sepsis model, and aid in designing exosome-based drug delivery systems for future therapeutic applications.

## Figures and Tables

**Figure 1 pharmaceutics-13-00427-f001:**
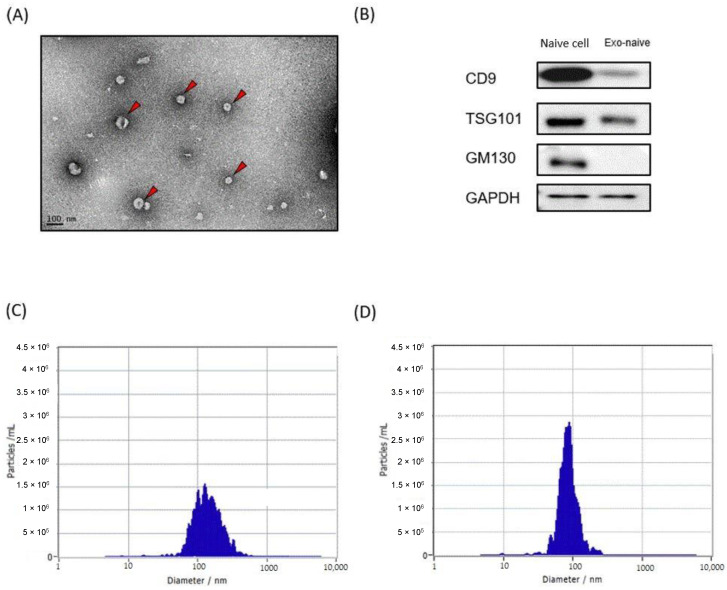
Exosome and liposome characterization. (**A**) Transmission electron microscopy of HEK293T-derived exosomes. (**B**) Western blot analysis of HEK293T cells and their exosomes for the indicated proteins. (**C**,**D**) Representative graphs of nanoparticle tracking analysis demonstrating size distribution and concentration of exosomes (**C**) and liposomes (**D**).

**Figure 2 pharmaceutics-13-00427-f002:**
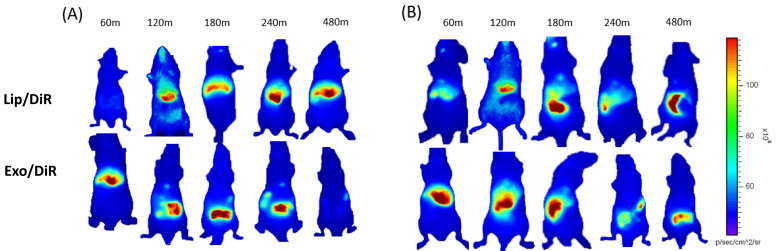
Whole-body NIR fluorescence imaging of mice after intravenous injection of HEK293T cell-derived exosomes and PEGylated liposomes. (**A**,**B**) NIR fluorescence images of healthy (**A**) and sepsis mice (**B**) taken 60, 120, 180, 240, and 480 min after intravenous injection of DiR-labeled liposomes and exosomes.

**Figure 3 pharmaceutics-13-00427-f003:**
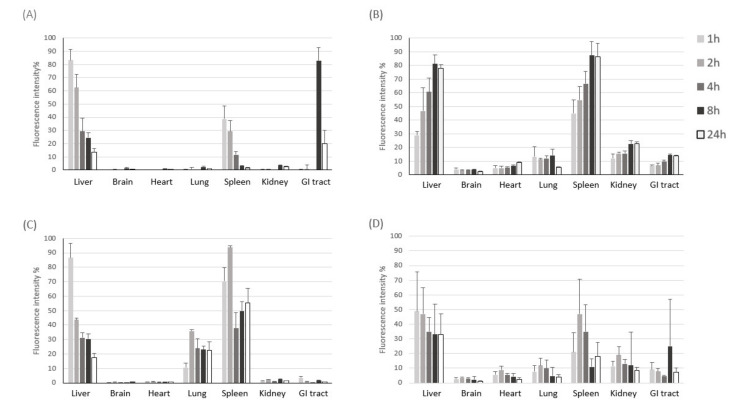
NIR fluorescence imaging of ex vivo organs after intravenous injection of HEK293T cell-derived exosomes and PEGylated liposomes. (**A**,**B**) Fluorescent signal intensity in each organ harvested at different time points after intravenous injection of DiR-labeled exosomes (**A**) and liposomes (**B**) into healthy mice. (**C**,**D**) Fluorescent signal intensity in each organ harvested at different time points after intravenous injection of DiR-labeled exosomes (**C**) and liposomes (**D**) into sepsis mice. The results represent mean ± SD (*n* = 5).

**Figure 4 pharmaceutics-13-00427-f004:**
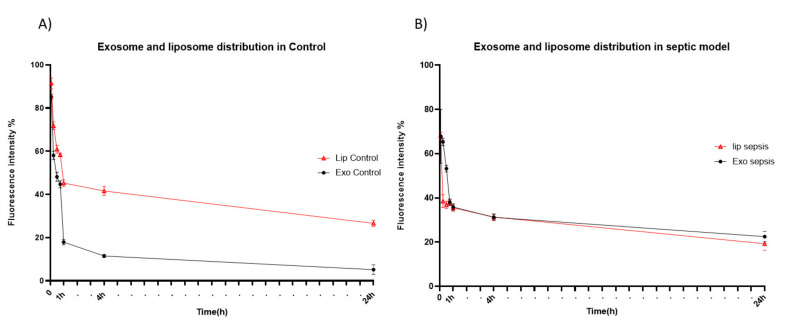
Blood circulation of HEK293T cell-derived exosomes and PEGylated liposomes after intravenous injection. (**A**,**B**) Fluorescent signal intensity of blood sample obtained from healthy (**A**) and sepsis mice (**B**) at different time points after intravenous injection of exosomes (closed circle) and liposomes (open triangle). The results represent mean ± SD (*n* = 5).

## Data Availability

Not applicable.

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
