# Peer review of "Biodistribution and Pharmacokinectics of Liposomes and Exosomes in a Mouse Model of Sepsis"

_pharmaceutics, 2021, doi:10.3390/pharmaceutics13030427_

Round 1

Reviewer 1 Report

The manuscript "Biodistribution and pharmacokinetics of liposomes and exosomes in a mouse model of sepsis" reports the 24 hours distribution of liposomes and exosomes labeled with a near infrared dye in a mouse model of sepsis. Although this study seems simple, it is very relevant and will greatly contribute in the advancement of exosomes as drug delivery carriers.

The manuscript is concise and well written.

It would have been more impactful to present a biodistribution study beyond the 24 hours and perhaps up to a month.

Author Response

Reviewer #1 (Remarks to the Authors) 

  1. It would have been more impactful to present a biodistribution study beyond the 24 hours and perhaps up to a month. 

- Thank you for this suggestion. It would have been interesting to explore this aspect. However, in the case of our study, it seems slightly out of scope because of the sepsis model's short lifetime.

Reviewer 2 Report

Dear Editor,

I have read with interest the manuscript presented by Dr. Choi, Dr. Park and collaborators, on the biodistribution and pharmacokinectics of liposomes and exosomes in a mouse model of sepsis.

The study is short and straightforward. The text is well written, and can be read quite easily, with high clarity. Authors did a good writing job and presented their work appropiately.

The experimental evidences support the claims. There are no detailed mechanistic explanations but this is acceptable as first report about observed behavior, and appears adequate to the scope of the article.

I have some major comments for the Authors. My request is that the authors briefly comment on them in the revised form of manuscript. Afterward, the manuscript can be accepted for publication.

Relevant Issues
1) Please double check in Figure 1B and Figure 1 caption whether the word "naive" is really correct or not.

2) Authors did not comment on the image of exosomes in healthy mouse (Figure 2A) at 480 minutes, when the signal disappeared. Please add a comment.

3) More comments should be added by referring to the fact that exosomes/liposomes were injected just after 30 minutes after LPS treatment. Why such a short time? Is this similar to other studies? What about injection after longer times? (is it experimentally feasible?) Please add details, and possibly add reference to previous studies to justify that choice.

4) In figure 4 the Authors should give an idea about the reproducibility of the curves. What is the experimental variability around each point? Error bars are not visible. If they are too small, please add a comment in the figure caption.

5) Part of liposomes also accumulate in the lung in sespis experiment (Figure 4D). Thus, the difference between liposomes and exosomes, based on this observation, is not very critical. Please comment on this aspect in the revised manuscript. In which sense exosomes behave differently in septic experiments (Figure 4C vs 4D)? Please explain.

6) Perhaps another striking difference between exosomes and liposomes is the time behavior of accumulation into organs. In figures 4A vs 4B the slope of time-dependent accumulation is opposite in the two cases. Exosomes have a rapid accumulation then disappear, whereas liposomes accumulate in time. Please comment on this aspect in the manuscript - if not yet done with due emphasis.

7) The final sentence "We believe that these sepsis-specific in vivo behaviors of exosomes can be utilized to develop exosome-based therapeutics for the treatment of sepsis and septic shock" needs more explanation. Why the Authors believe so. What is the scientific basis for this opinion. What are the hints obtained by this study and in which manner this study can provide information, and in which direction. Please comment on this important point that is the major conclusion of the study. How this study helps for future developments?

Minor/Typos
1) Line 115; please add "C" after the degree symbol
2) Line 146; please use the micro (μ) symbol for 200 uL

Ethical considerations
The Authors declared that they have carried out their research according to the protocols of the Institutional Animal Care and Use Committee (IACUC) of KAIST (Korea advanced institute of science and technology). 

Author Response

Reviewer #2 (Remarks to the Authors) 

  1. Please double check in Figure 1B and Figure 1 caption whether the word "naive" is really correct or not.

- Thanks for your valuable comment. We have removed the mentioned word from Fig. 1.

In Figure 1,

Figure 1. Exosome and liposome characterization. (A) Transmission electron microscopy of HEK293T-derived exosomes. (B) Western blot analysis of HEK293T cells and their exosomes for the indicated proteins. (C and D) Representative graphs of nanoparticle tracking analysis demonstrating size distribution and concentration of exosomes (C) and liposomes (D).

  1. Authors did not comment on the image of exosomes in healthy mouse (Figure 2A) at 480 minutes, when the signal disappeared. Please add a comment.

- Thank you for your valuable comment. We believe due to the nature of lipophilic dyes, the disappearance of the signal at 480 minutes might be because of excretion of the dye from the GI track. The weak signal can be seen by zooming to the GI area. We have added a sentence related to this result in the main text.

In the main text (page 6, lines 201-204)

“Interestingly, exosomes accumulated in the healthy liver translocated to the intestine from 2 hours post-injection and almost disappeared within the 8 hours while the sepsis liver somewhat delayed translocation of exosomes to the intestine.”

  1. More comments should be added by referring to the fact that exosomes/liposomes were injected just after 30 minutes after LPS treatment. Why such a short time? Is this similar to other studies? What about injection after longer times? (is it experimentally feasible?) Please add details, and possibly add reference to previous studies to justify that choice.

- Thank you for your valuable comment. The toxic issues of LPS are mediated by cytokines like TNF-α, interleukin 1 (IL-1), IL-6 released by LPS-activated host cells, such as macrophages and neutrophils (doi.org/10.1371/journal.pone.0106331). Several studies have shown that initial pro-inflammatory cytokines increase rapidly within 30 minutes after LPS injection (dx.doi.org/10.1128%2FCDLI.11.3.452-457.2004, doi.org/10.1186/s12950-018-0197-4). Our recent cellular biodistribution findings also confirm the significant therapeutic effect of exosomes carrying srIKB during the early stage of sepsis before the immunosuppressive state (DOI: 10.1126/sciadv.aaz6980). Therefore, authors believe the early time points would be crucial for a better understanding of distribution behavior. We have added a sentence and references related to this experimental setting in the main text.

In the main text (page 5, lines 184-191)

“The toxic issues of LPS are mediated by cytokines like TNF-α, interleukin 1(IL-1), IL-6 released by LPS-activated host cells, such as neutrophils and macrophages. Several studies have shown that initial pro-inflammatory cytokines increase rapidly within 30 minutes after LPS injection [31-32]. Our recent cellular biodistribution findings also confirm the significant therapeutic effect of exosome carrying srIKB during the early stage of sepsis before the immunosuppressive state [30]. Therefore, in this study, exosome and liposome were injected 30 minutes after LPS injection to investigate their biodistribution in the early stage of sepsis.”

In References,

“31. S. Miyazaki, F. Ishikawa, T. Fujikawa, S. Nagata, K. Yamaguchi, Intraperitoneal Injection of Lipopolysaccharide Induces Dynamic Migration of Gr-1high Polymorphonuclear Neutrophils in the Murine Abdominal Cavity, Clinical and Diagnostic Laboratory Immunology, 11 (2004) 452.

  1. H. Fang, A. Liu, X. Chen, W. Cheng, O. Dirsch, and U. Dahmen, The severity of LPS induced inflammatory injury is negatively associated with the functional liver mass after LPS injection in rat model. Journal of Inflammation, 15 (2018) 21.”

  1. In figure 4 the Authors should give an idea about the reproducibility of the curves. What is the experimental variability around each point? Error bars are not visible. If they are too small, please add a comment in the figure caption.

- Thanks for your valuable comment and sorry for our mistake. I think the reviewer meant Figure 3, not Figure 4. We have added the error bars in the graph.

In Figure 4,

  1. Part of liposomes also accumulate in the lung in sespis experiment (Figure 4D). Thus, the difference between liposomes and exosomes, based on this observation, is not very critical. Please comment on this aspect in the revised manuscript. In which sense exosomes behave differently in septic experiments (Figure 4C vs 4D)? Please explain.

- Thank you for your valuable comment. The amount of liposomes accumulated in the sepsis lung was similar to that in the control lung (Figures 3B and 3D), which was different from the sepsis-specific pattern observed with exosomes. We have added a sentence related to this result in the main text.

In the main text (page 6, lines 227-228)

“Liposomes did not show sepsis-specific accumulation in the lung (Figs. 3B and 3D).”

  1. Perhaps another striking difference between exosomes and liposomes is the time behavior of accumulation into organs. In figures 4A vs 4B the slope of time-dependent accumulation is opposite in the two cases. Exosomes have a rapid accumulation then disappear, whereas liposomes accumulate in time. Please comment on this aspect in the manuscript - if not yet done with due emphasis.

- Thank you for your valuable comment. As suggested, we have revised the manuscript accordingly.

In the main text (page 6, lines 218-222)

“On the other hand, the covalent linking of polyethylene glycol (PEG) chains is an effective method to prolong the blood circulation time of liposomes. The resistance of protein opsonins absorbance onto liposome surfaces caused by PEGylation delayed their elimination by the mononuclear phagocytic cells in the liver and spleen (Fig. 3B).”

  1. The final sentence "We believe that these sepsis-specific in vivo behaviors of exosomes can be utilized to develop exosome-based therapeutics for the treatment of sepsis and septic shock" needs more explanation. Why the Authors believe so. What is the scientific basis for this opinion. What are the hints obtained by this study and in which manner this study can provide information, and in which direction. Please comment on this important point that is the major conclusion of the study. How this study helps for future developments?

- Thank you for your valuable comment. Numerous studies showed the potential exosome-related pathogeneses and therapies for sepsis, ARDS, and ALI treatment ( doi.org/10.1016/j.biopha.2019.108748, DOI: 10.1126/sciadv.aaz6980). The authors believe that these findings might help better understand the exosome organ distribution, particularly in the sepsis model, and can aid in designing exosome-based nanocarriers for future applications. Accordingly, the conclusion has been revised to emphasize this point.

In the main text (page 8, lines 253-256)

“Thus, these findings might help better understanding of the exosome and liposome organ distribution, particularly in the sepsis model, and can aid in designing exosome-based drug delivery systems for future therapeutic applications.”

  1. Minor/Typos

1) Line 115; please add "C" after the degree symbol

2) Line 146; please use the micro (μ) symbol for 200 uL

- Thank you for your valuable comment. All above typos have been corrected throughout the manuscript.
